# Integrated model of the vertebrate augmin complex

Sophie M. Travis[1,7], Brian P. Mahon [1,4,7], Wei Huang [2], Meisheng Ma[3,5], Michael J. Rale [1,6], Jodi Kraus[1], Derek J. Taylor [2], Rui Zhang [3] & Sabine Petry [1]

Accurate segregation of chromosomes is required to maintain genome integrity during cell division. This feat is accomplished by the microtubule-based spindle. To build a spindle rapidly and with high fidelity, cells take advantage of branching microtubule nucleation, which rapidly amplifies microtubules during cell division. Branching microtubule nucleation relies on the hetero-octameric augmin complex, but lack of structure information about augmin has hindered understanding how it promotes branching. In this work, we combine cryo-electron microscopy, protein structural prediction, and visualization of fused bulky tags via negative stain electron microscopy to identify the location and orientation of each subunit within the augmin structure. Evolutionary analysis shows that augmin's structure is highly conserved across eukaryotes, and that augmin contains a previously unidentified microtubule binding site. Thus, our findings provide insight into the mechanism of branching microtubule nucleation.

Spindles are micron-scale microtubule (MT)-based structures that are built from scratch every time a eukaryotic cell divides. Spindle MTs are generated in a manner that is tightly spatially controlled, with their stable minus-ends pointing toward the spindle pole on the cell periphery and their growing plus-ends pointing toward the chromosomes at the spindle center. Critically, spindle MTs must be continuously generated, as any given MT turns over within seconds, yet the spindle framework must often persist for up to an hour[1,2]. Due to the spindle's central importance for viability, multiple partially redundant pathways are used to generate spindle MTs. Spindle MTs have been shown to originate from centrosomes[3,4] and kinetochores[5,6], while evidence has emerged that branching MT nucleation is responsible for generating the bulk of the MTs in the spindle[7–9].

In branching MT nucleation, a new MT is nucleated from a side of an existing MT at a shallow angle[10–12]. Thus, MTs can be rapidly amplified while preserving their polarity. In the absence of branching, spindle formation is delayed, spindle density decreases, mitosis stalls, and cytokinesis cannot proceed[13–16]. Branching MT nucleation depends on recruitment of the MT template and nucleator, the γ-tubulin ring complex (γ-TuRC), to the side of a pre-existing mother MT to give rise to a branched MT with matching polarity to the mother[12,17–19]. Both recruitment of γ-TuRC, as well as maintenance of polarity[17–20], are carried out by a conserved protein complex, augmin.

Augmin is an eight-subunit protein complex first discovered for its role in localizing γ-TuRC to cellular spindles. It was discovered independently in invertebrates[13], vertebrates[14,21], and plants[22]. Vertebrate augmin consists of eight subunits, Haus1-8, that form two biochemically distinct subcomplexes, tetramers T-II and T-III. T-II binds MTs via the disordered N-terminus of Haus8[20,23], whereas T-III binds to γ-TuRC[20] via the adaptor subunit NEDD1[19], thus forming a bridge between the mother MT and the source of the branched MT. Negative stain electron microscopy established that the complex from both

[1]Department of Molecular Biology, Princeton University, Princeton, NJ, USA. [2]Department of Pharmacology, Case Western Reserve University, Cleveland, OH, USA. [3]Department of Biochemistry and Molecular Biophysics, Washington University in St. Louis, School of Medicine, St. Louis, MO, USA. [4]Present address: Department of Structural Biology, Bristol Myers Squibb, Princeton, NJ, USA. [5]Present address: Department of Histology and Embryology, School of Basic Medicine, Tongji Medical College, Huazhong University of Science and Technology, Wuhan, Hubei, China. [6]Present address: Department of Cell Biology, Harvard Medical School, Boston, MA, USA. [7]These authors contributed equally: Sophie M Travis, Brian P Mahon. ✉e-mail: zhangrui@wustl.edu; spetry@princeton.edu

human and *X. laevis* forms an "h"-shape, with the bottom of the "h" sitting on an MT, and the stalk region pointing away[20,23]. However, to date, no high resolution three-dimensional structural information has been available for either the complex as a whole or any fragments. In addition, the lack of any identified structural homologs precludes hypotheses about augmin's subunits and their organization within the complex, and, finally, it remains unclear how augmin interacts with either the MT or γ-TuRC to enable branching MT nucleation.

In this work, we report a medium resolution cryo-electron microscopy (cryo-EM) map of the *X. laevis* augmin complex. By leveraging recent advances in protein structural prediction, as implemented in AlphaFold2-Multimer, and resolving structural ambiguities by tagging experiments coupled with negative stain electron microscopy, we report a molecular model of *X. laevis* augmin. Integrating this model with prior experiments, we identified the locations of the MT binding site, a putative secondary MT binding site, and the proposed γ-TuRC binding site, and demonstrated that the structure of augmin is highly conserved across eukaryotes. These results allow us to build a molecular model of the branch site, and, in addition, reveal insight into the shared evolutionary origin of augmin and the NDC80 kinetochore-localized complex.

## Results

### Cryo-EM study of *X. laevis* augmin

The augmin complex is a flexible and extended assembly, and this dynamic nature makes augmin's structure determination challenging. As the most suitable target for structural studies, we chose *X. laevis* augmin complex lacking the unstructured, poorly conserved 548 residues of Haus6's C-terminus, a truncated complex which we had previously reconstituted and purified[20]. Having recapitulated this purification strategy (Fig. 1a), we next assessed augmin's activity. Although the C-terminus of Haus6 has been predicted to bind to γ-TuRC[21], and thus contributes to augmin's branching activity, we had previously shown that truncated augmin retains the ability bind γ-TuRC via T-III, as well as the ability to bind MTs, and can partially substitute for full-length Haus6 in supporting branching MT nucleation in extract[20].

Thus, to determine the biochemical activity of our cryo-EM sample, we reconstituted branching MT nucleation in vitro using a system containing *X. laevis* augmin lacking the C-terminus of Haus6 and γ-TuRC purified using its activator, the γ-TuNA domain of CDK5RAP2[24]. We first incubated augmin and γ-TuRC with GMPCPP-stabilized MT seeds attached to glass to selectively localize these proteins to the seeds prior to adding tubulin in a reaction buffer containing GTP. We indeed observed localization of the GFP-tagged augmin complex to the MT seeds. Moreover, upon addition of tubulin, new MTs nucleated from the side of the mother MT. Because the augmin complex has been previously demonstrated to have no intrinsic ability to nucleate MTs[17–19], these results suggest that our purified *X. laevis* augmin is capable of recruiting γ-TuRC to the MT seeds and, additionally, of stimulating formation of new MT branches. Thus, augmin lacking the C-terminus of Haus6 is biochemically active in its three main functions, namely binding MTs, recruiting γ-TuRC, and facilitating branching MT nucleation (Fig. 1b, Supplementary Fig. 1). In addition, we find that truncated augmin can support branching MT nucleation in the absence of TPX2, a branching factor absolutely required in *Xenopus* egg extract. This result is consistent with previous findings that full-length vertebrate augmin is sufficient, with γ-TuRC, for MT branching in vitro[17,19], although we have also previously shown that, for *Xenopus* proteins, TPX2 dramatically enhances branching activity in vitro[17].

Once we had confirmed complex activity, we employed single-particle cryo-EM to determine the structure of the augmin complex. We optimized the cryo-EM sample freezing condition and found that Quantifoil holey carbon grids covered with additional continuous thin carbon film (~5 nm thickness) produced the best mono-dispersed

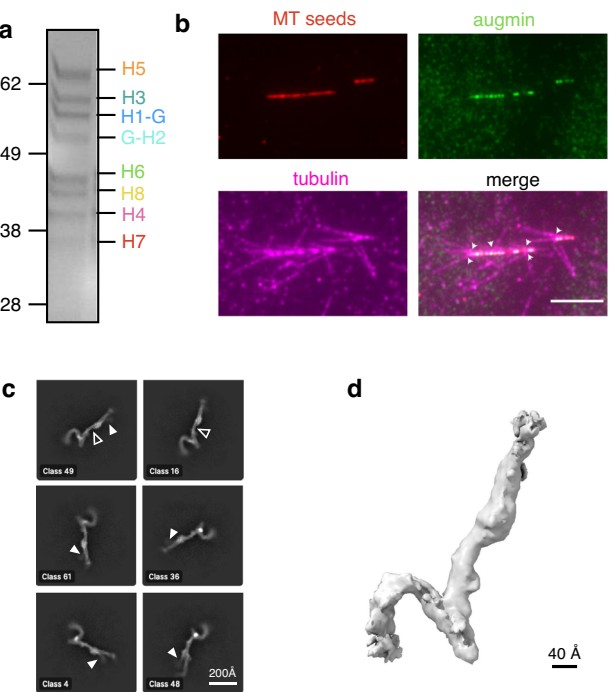

**Fig. 1 | Cryo-electron microscopy structure of the *X. laevis* augmin complex.** **a** Stoichiometric *X. laevis* augmin octameric complex (lacking the C-terminus of Haus6) purified from insect cells. Here, and in subsequent figures, Haus1 through H8 is abbreviated as H1 through H8. Molecular weight standards are included at left, in units of kDa. Representative image is shown out of 5 biological replicates. **b** Representative Total Internal Reflection Fluorescence (TIRF) microscopy images from in vitro branching reconstitution with augmin and γ-TuRC. Alexa-568 labeled stabilized microtubule (MT) seeds (red, top) were mixed with GFP-augmin and γ-TuRC allowing for localization of all components to the mother MT (green, 2nd panel) prior to initiation of in vitro branching MT nucleation. Branching MT nucleation was initiated by introducing 20 μM tubulin labeled with Cy5 (purple, bottom) whereby it was recruited to the stabilized MT seeds (purple, bottom), and formed new MTs at an acute branch angle (arrowheads). The field of view has been cropped for clarity to show branching MT nucleation at one template MT. The entire field of view can be found in Supplementary Fig. 1b. Reactions were performed at 33 °C. Scale bars correspond to 5 μm. Two technical replicates were performed. **c** 2D class averages of the augmin complex, demonstrating density patterns consistent with secondary structure (open arrows). Some classes, particularly Class 48 (bottom right), show faint density for a second "leg" in T-III (solid arrows). **d** Sharpened electron density map of the augmin complex. Black scale bar indicates a diameter measurement of 40 Å.

augmin particles. Using this strategy, we collected ~3000 micrographs containing intact augmin particles. Due to the background noise of the added carbon, along with augmin being an inherently elongated particle, we encountered low particle contrast within our micrographs, which complicated data processing. Therefore, we used multiple strategies for particle picking, beginning with templates derived from negative stain data. Following 2D classification of the template picking results, which displayed severe preferred orientation bias, we used the best particles to generate an initial 3D model of the augmin complex, create new, evenly spaced templates, and repick ~100,000 intact augmin particles, resulting in 2D classes corresponding to additional views and displaying secondary structure features (Fig. 1c, Supplementary Fig. 2a).

Next, we performed 3D reconstruction and refinement to obtain a cryo-EM reconstruction at 6.9 Å resolution (Fig. 1d, Supplementary Fig. 2b, Supplementary Table 1). The map displayed residual anisotropy due to uneven angular distribution of the augmin particles on the EM grid (Supplementary Fig. 2c) and, although in the preferred orientations of the map, secondary structure elements were

visible, the overall map did not display interpretable secondary structure elements. Thus, we believe that the resolution estimated by CryoSPARC is higher than the effective resolution of the map. However, we were able to establish unambiguously that augmin has a relatively uniform radius across its entire length, comprising a cylindrical density of ~40 Å in diameter (Fig. 1d). This radius is consistent with that of a four-helix bundle, but, as even 6.9 Å resolution would be insufficient for de novo model building, we needed to incorporate structural information from orthogonal sources to further interpret the cryo-EM density.

### Integrated structural model of *X. laevis* augmin

The challenge of interpreting moderate-resolution density is an old one in structural biology, and much insight has been gained by docking high-resolution structural fragments or orthologous structures into low resolution maps (for example in the interpretation of the COPI vesicle coat[25]). However, this approach was not feasible for augmin because no high-resolution subcomplexes had previously been determined, nor had any structural homologues been identified. An unexpected solution presented itself last year with the groundbreaking progress in protein structural prediction heralded by the release of AlphaFold2[26] and, particularly, the updated algorithm optimized for multi-subunit complexes, AlphaFold2-Multimer[27] (as, for example recently implemented in interpretation of the nuclear pore complex[28]).

Although the first iteration of AlphaFold2, which can model only individual subunits, predicted extended helical structures that were incompatible with the 6.9 Å map, we were able to gain much more insight from the updated AlphaFold2-Multimer release[27]. Leveraging our knowledge that augmin can be divided into two tetrameric subcomplexes—T-II, comprising Haus2, and Haus6-8, and T-III, comprising Haus1, and Haus3-5—we were able to predict structures of these two stable tetramers (Fig. 2a; Supplementary Fig. 3a). The predicted models of T-II and T-III had high coordinate confidence (as assessed by predicted local distance difference test; pLDDT[29]) across the majority of the structure (Supplementary Fig. 3a), and the five independent models predicted for each subcomplex agreed well in their overall fold and conformation (Supplementary Fig. 3a). The models primarily diverged in poorly predicted regions with low

pLDDT, such as the disordered N-terminal MT binding domain (MTBD) of Haus8, but three high-confidence regions also differed: a second leg emerging from the bulge of T-III, the base of T-III, and the globular domains at the end of T-II (Supplementary Fig. 3b). To distinguish which predicted conformation best fit the cryo-EM map, we rigid-body fit the models into the map and determined the cross-correlation between predicted and observed electron density. In general, all models corresponded relatively well to the electron density, but we obtained the best fit for T-II model 5 and T-III model 2 (Supplementary Fig. 3b).

In general, the predicted T-II and T-III structures were in good agreement with the cryo-EM map, forming extended four-helix bundle structures that match the radius observed in the map (Fig. 2a). AlphaFold2 was also able to correctly predict a a helical hairpin inserted in Haus5 (Fig. 2b) and a set of insertions in the center of T-III matching the bulge observed in the center of the T-III stalk (Fig. 2c). In addition to this broad agreement with the 6.9 Å cryo-EM map, the AlphaFold2-Multimer models recapitulated the predicted helical and entwined nature of subunits within the tetramers and were consistent with the assembly hierarchy previously determined by pull-downs and mass spectrometry[20].

Two main regions of the augmin model did not fit within the contours of our moderate resolution map: the second leg of T-III (Fig. 2d) and the N-terminal globular domains of T-II (Fig. 2e). These regions also corresponded to the regions of greatest conformational uncertainty as predicted by AlphaFold2 (Supplementary Fig. 3b). The first low-confidence region, the T-III second leg, appears as a faint feature in some 2D class averages (Fig. 1c); however, we see little evidence of electron density corresponding to the second leg in our 3D maps (Fig. 2d). This suggests that the second leg may be relatively flexible. The two globular domains at the extreme tip of T-II also fit our map poorly, but for the opposite reason: the electron density is larger than can be explained by the calponin homology domains (Fig. 2e). As this region corresponds to one extreme end of the particle, it is possible that rotational uncertainty during particle alignment may result in loss of resolution and smearing out of the map. However, this mismatch could also herald dynamic motion in this region of augmin, in agreement with the conformational variability present in the AlphaFold2 predictions.

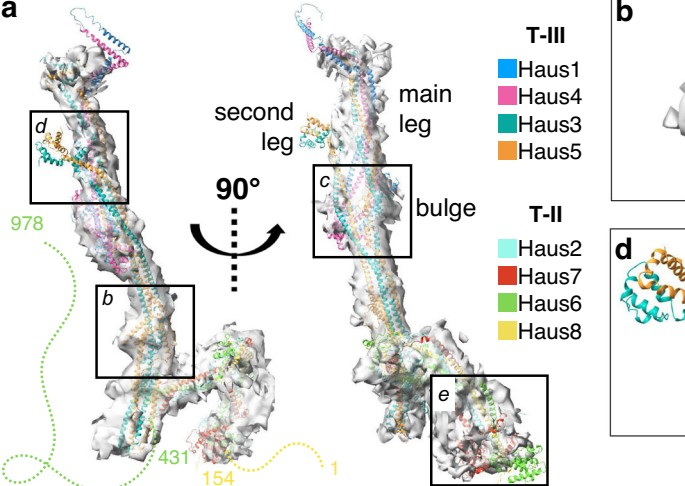

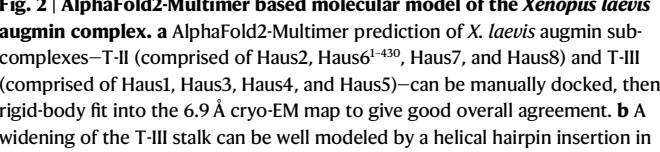

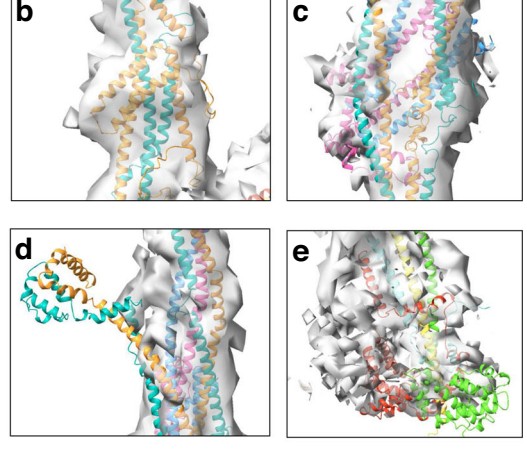

**Fig. 2 | AlphaFold2-Multimer based molecular model of the *Xenopus laevis* augmin complex. a** AlphaFold2-Multimer prediction of *X. laevis* augmin subcomplexes—T-II (comprised of Haus2, Haus6[1–430], Haus7, and Haus8) and T-III (comprised of Haus1, Haus3, Haus4, and Haus5)—can be manually docked, then rigid-body fit into the 6.9 Å cryo-EM map to give good overall agreement. **b** A widening of the T-III stalk can be well modeled by a helical hairpin insertion in Haus5. **c** A bulge halfway down the T-III stalk can be modeled by α-helical bundles predicted at the N-termini of Haus1 and Haus4. **d** The second leg of T-III, comprising the extreme N-terminus of Haus3 and Haus5, has no corresponding density in the cryo-EM map, perhaps due to flexibility. **e** The globular domains at the N-termini of Haus6 and Haus7 imperfectly fit the cryo-EM map, perhaps a result of multiple conformational states.

The structure of augmin, as guided by prediction by AlphaFold2-Multimer, provided insight into augmin's global architecture. Perhaps unexpectedly, although T-II and T-III both consist of a core parallel 4-helix bundle and each is composed of a dimer of dimers, the global architecture of the two tetramers differs substantially. T-III[core] is composed of paralogous subunits Haus1 and Haus4, which form a coiled-coil structure and intertwine with the coiled-coil C-termini of a second set of paralogs, Haus3 and Haus5 (Supplementary Fig. 4a, b). The remaining three quarters of Haus3 and Haus5 form a continuous coiled-coil that folds back on itself into an extended antiparallel four-helix bundle, forming T-III[ext], a bundle of equivalent length to T-III[core] (Supplementary Fig. 4a, b). T-II is also comprised of two sets of paralogues, Haus6/7 and Haus2/8. However, in contrast with T-III[core], in T-II paralogs do not form coiled coils with one another, but rather Haus6 and Haus8—structurally unrelated to one another—form a tight dimer that then associates with the Haus2/Haus7 dimer (Supplementary Fig. 4a, c). Not only does T-III not share homology with T-II, but, following a DALI homology search[30] of the AlphaFold2 predicted human proteome[31], no obvious structural homologues could be found for any T-III subunits. However, the two T-II subunits Haus6 and Haus7 were unexpectedly predicted to contain globular calponin homology (CH) domains at their N-termini (Supplementary Table 2), which will be discussed further below.

## Validation with negative stain electron microscopy

Because structural prediction using AlphaFold2 is a comparatively recent development, we wanted to test the proposed augmin model using an established orthogonal technique that has previously been used to interpret the architecture of low-resolution electron microscopy structures[32,33]. To independently establish the position and orientation of each subunit within T-III, we fused a bulky GFP tag, attached by a minimal linker, at the N- or C-terminus of each subunit of the tetramer. After purifying the tagged augmin tetramer complexes, as well as a wholly untagged complex, we performed negative stain electron microscopy. As previously reported[20], the structure of T-III on its own recapitulates the stalk region found in intact augmin (Fig. 3a). Therefore, we introduced C-terminal GFP tags to each of the four subunits and looked for additional density absent from the untagged complex, which we reasoned would correspond to the GFP. We identified a single additional globular

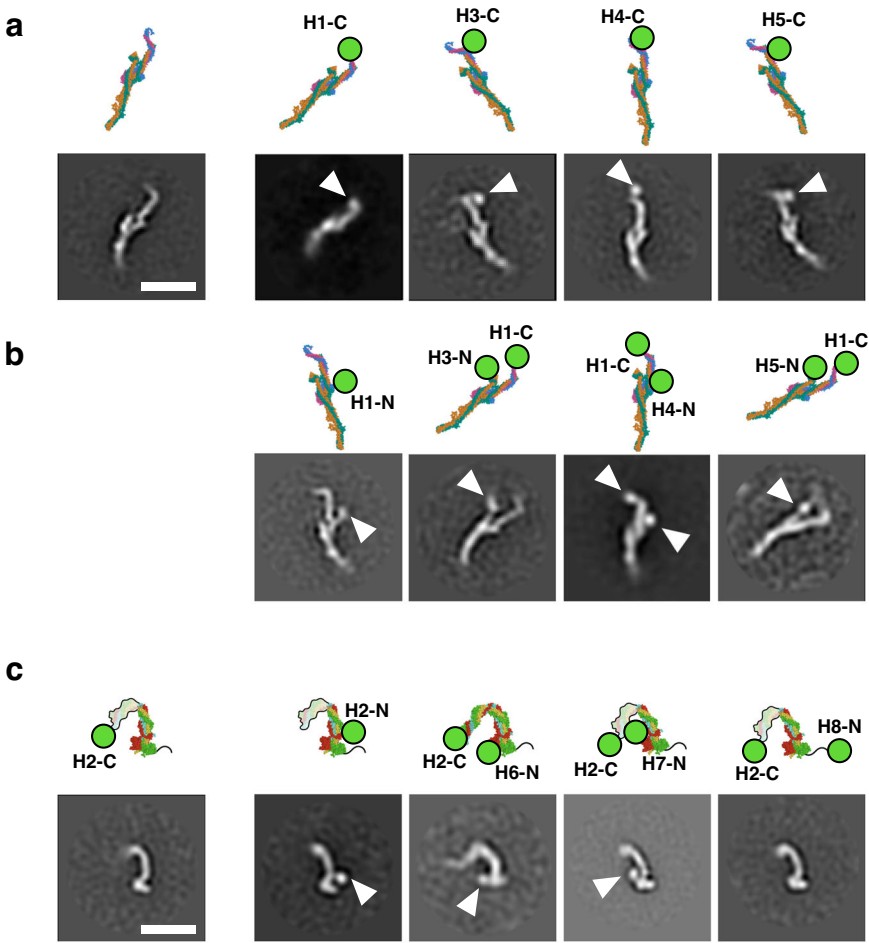

**Fig. 3 | Negative staining EM validation of augmin model. a** Five T-III subcomplexes untagged or singly tagged with GFP at the C-termini of subunits. Top, the molecular model of T-III is displayed, rotated to correspond to the corresponding 2D class below. Positions of any GFP tags are indicated with a green circle and the tagged subunit and terminus are indicated (e.g., T-III singly tagged with a GFP at the C-terminus of Haus1 is designated as H1-C). Bottom, representative 2D negative stain classes are displayed for each differently tagged complex. Additional density found in the tagged complexes, but not the untagged control at left, is indicated by solid arrows. Scale bar is 200 Å. **b** Four additional T-III subcomplexes singly or doubly tagged with GFP. **c** Five T-II subcomplexes singly or doubly tagged with GFP. As in Fig. 3a, molecular models of T-II are displayed in the top row, and experimentally observed 2D classes in the bottom. Models are oriented similarly to the 2D class below and position(s) of GFP tags are indicated with green circles. Additional density found in the tagged complexes, but not the untagged control at left, is indicated by solid arrows. The C-terminal half of T-II is disordered in most 2D classes, and this missing density is indicated in the molecular model by an outlined pastel region. Similarly, the unstructured N-terminus of Haus8 is indicated by a black squiggle. Scale bar is 200 Å.

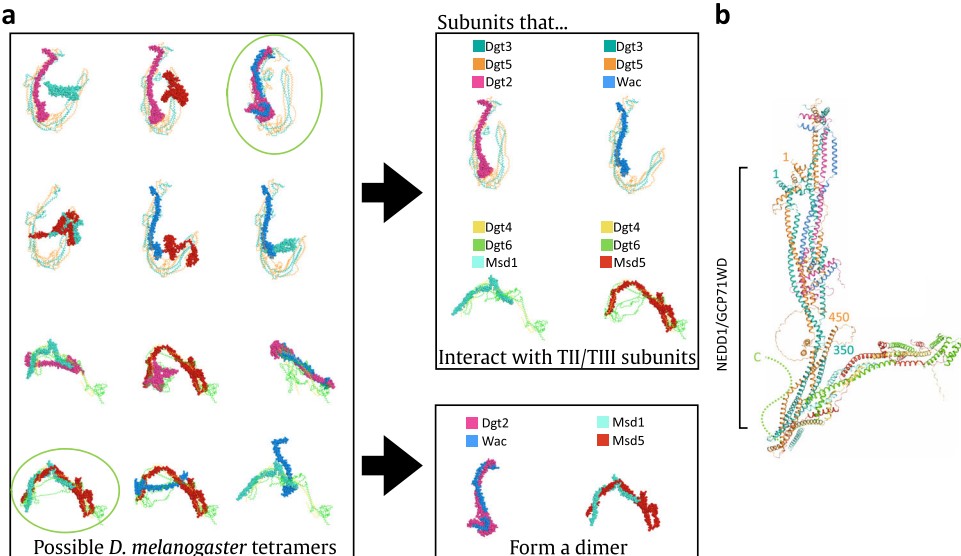

**Fig. 4 | Homology-based model of the *D. melanogaster* augmin complex. a** The structures of all six possible *D. melanogaster* T-III subcomplexes (top) and possible T-II subcomplexes (bottom) were predicted by AlphaFold2-Multimer. Structures that superimpose with *X. laevis* T-II or T-III are circled. By categorizing the resulting predictions, Dgt2 and Wac were found to bind consistently with T-III subunits Dgt3 and Dgt5, as well as dimerizing with one another, and, similarly, Msd1 and Msd5 consistently dimerized with one another and bound T-II subunits Dgt4 and Dgt6. **b** Structural alignment of the *D. melanogaster* subcomplexes T-II and T-III resulted in a molecular model of *D. melanogaster* augmin. The γ-TuRC adaptor protein GCP71WD (aka NEDD1)[35], has been demonstrated to bind to the heterodimer of Dgt3$^{1-350}$ and Dgt5$^{1-450}$, and that span of residues, comprising the likely NEDD1 binding site, have been annotated.

density on each singly tagged complex, located at the bent end of the T-III stalk (Fig. 3a). This confirmed that the C-termini of all four T-III subunits localize to the end of the stalk, adjacent to one another (Fig. 3a).

Next, to establish the orientation of subunits within T-III, N-terminal GFP tags were introduced to the subunits. In most cases, a second, C-terminal GFP tag was introduced to Haus1 to aid in purification. Consistent with the AlphaFold2-Multimer model, the N-termini of Haus1 and Haus4 localized near the center of the stalk, above the bulge (Fig. 3b). Additional density consistent with the N-termini of Haus3 and Haus5 could be visualized protruding from a second leg, on the opposite side of the rod from Haus1 and Haus5 (Fig. 3b). As discussed in the preceding section, this second leg is visible in select cryo-EM 2D classes (Fig. 1c) but is missing from the finalized 3D cryo-EM map, and we had interpreted these observations to suggest this region was flexible. Similarly, in the negative stain 2D classes, the leg was only visible in a small subset of classes, further confirming that the extreme N-terminal regions of Haus3 and Haus5 are likely to be dynamic.

We performed similar experiments to confirm the location of subunits within T-II. Unlike T-III, the majority of T-II 2D classes recapitulated only half of the arch structure observed in the T-II region of intact augmin (Fig. 3c). The visible half of the T-II arch contains two globular domains, and thus corresponds to the region of T-II furthest from T-III and is predicted to contain the N-termini of T-II subunits. In agreement with this assignment, incorporation of N-terminal GFP tags on Haus2, Haus6, and Haus7 yielded additional density at the globular-domain containing end of the tetramer (Fig. 3c). As the extreme N-terminus of Haus8, which contains augmin's primary MT binding site[23,34], is predicted to be disordered, we were not surprised to find that introducing a GFP tag to the N-terminus of Haus8 yielded no additional observed density. Similarly, we did not observe additional density when a GFP tag was introduced to the C-terminus of Haus2, due to the complete lack of density corresponding to that region of the T-II arch. In sum, for both T-II and T-III, visualization of fused bulky tags via negative stain electron microscopy helped validate our augmin model.

## Structural conservation of augmin across diverse eukaryotes
As discussed above, the augmin complex was first discovered in invertebrates (*D. melanogaster*)[13], and only afterwards identified in vertebrates (*H. sapiens*)[14] and plants (*A. thaliana*)[22]. Surprisingly, although a 1:1 correspondence has been established between vertebrate and plant augmin subunits—where for example Haus1 in vertebrates is equivalent to Aug1 in plants, Haus2 to Aug2, and so on—only 4 out of the 8 insect augmin subunits could be matched with their much more closely related vertebrate counterparts. For insects, the large subunits Dgt3 and Dgt5 have clear homology to Haus3 and Haus5, respectively, and Dgt6 and Dgt4 to Haus6 and Haus8. However, it was unclear to which vertebrate subunit the remaining four small insect subunits corresponded to, or even which tetramer they belonged in. Using AlphaFold2-Multimer to predict the structure of *D. melanogaster* augmin, we were able to resolve this conundrum. We ran combinatorial predictions of all 12 possible *D. melanogaster* tetramers, containing either Dgt3/5 or Dgt4/6 and each possible pair of unassigned small subunits (Fig. 4a). Only two predictions yielded solutions with structural homology to vertebrate augmin and thus we were able to equate Dgt2 with Haus4 (T-III), Dgt8 with Haus1 (T-III), Msd1 with Haus2 (T-II), and Msd5 with Haus7 (T-II).

Despite the low sequence similarity between *X. laevis* and *D. melanogaster* subunits, the predicted structure of *D. melanogaster* augmin well matched that of *X. laevis* augmin, particularly in the T-III subcomplex (Fig. 4b). T-II displayed the same fold between both species, but *D. melanogaster* T-II was predicted to form a wider and shallower arch, due to a relative extension of its hinge. To further validate the *D. melanogaster* model, we integrated structural restraints previously obtained by the Wakefield group through cross-linking mass spectrometry (XLMS)[35]. By incorporating these data into our structural model, we were able to demonstrate that AlphaFold2-Multimer's prediction of *D. melanogaster* augmin matched the majority of XLMS restraints, with the major exception of the only high-confidence cross-tetramer crosslink, between Msd1 Lys-113 and Dgt3 Ser-165 (Supplementary Fig. 5, Supplementary Table 3). However, as the C-termini of all four *D. melanogaster* T-II subunits are also substantially shorter than their vertebrate counterparts, it seems likely that the interface

between the two tetramers has coevolved to diverge from that found in vertebrate augmin.

In addition, with the *D. melanogaster* augmin model in hand, we were able to integrate further biochemical data about the location of binding sites on augmin for the γ-TuRC nucleator. Previous experiments have established the binding site of the γ-TuRC adaptor NEDD1−GCP71WD in *D. melanogaster*[35]. Chen and colleagues not only purified a heterodimer consisting of Dgt3$^{1-350}$ and Dgt5$^{1-450}$—roughly equivalent to T-III$^{ext}$—they also demonstrated that this dimer was sufficient to bind to GCP71WD. Thus, we can put these facts together to map the NEDD1 binding location on *D. melanogaster* augmin to T-III$^{ext}$ (Fig. 4b). Intriguingly, this putative γ-TuRC binding location is adjacent to the unstructured C-terminus of Haus6, which, in vertebrates, has been proposed as the binding site of NEDD1[21,36]. Therefore, regardless of the organism, γ-TuRC is likely to bind to the base of the T-III stalk.

Expanding upon our results with *D. melanogaster* augmin, we next sought to answer another major question about the augmin complex, namely how well conserved it is across eukaryotes. Although no comprehensive analysis of augmin conservation is available, the complex has been reported missing from both budding and fission yeast, as well as nematodes[37]. To begin with, we performed a sequence-level search for homologs of the best-conserved subunit, Haus6, across the eukaryotic kingdom, and found that Haus6 was present in all five eukaryotic supergroups, although only well-conserved in metazoans and plants (Supplementary Fig. 6a). It remained unclear, however, whether the structure of the full complex was also conserved. Thus, we extended our prediction into representative members of each supergroup, except for the SAR supergroup (*Stramenopiles/Alveolata/Rhizaria*) where only Haus6 could be reliably identified in a single key species, *Tetrahymena thermophila*. Although, as with *Drosophila*, some small augmin subunits in these divergent eukaryotes could not be identified by sequence homology, structural prediction using all identified orthologues resulted in similar structures across *Opisthokonta*, *Plantae*, *Amoebozoa*, and *Discoba*, even among species with unidentified subunits (Supplementary Fig. 6b). This suggests that cryptic orthologs of the remaining small augmin subunits are likely present in these recalcitrant genomes and may perhaps be identifiable in the future using a 3D-structure-based search strategy such as DALI[30] coupled with AlphaFold2 prediction of individual subunits. In addition, the augmin complex was apparently present in the last eukaryotic common ancestor and, although the full complex has been lost in many species since, where retained, augmin retains its structure, and thus perhaps its function.

## Discussion

In this work, we have combined three orthogonal techniques—medium resolution cryo-EM, structural prediction using AlphaFold2-Multimer, and subunit/subcomplex localization using negative stain electron microscopy—to assemble a molecular model of the *X. laevis* augmin complex. This model has enabled us not only to identify the positions and orientations of all eight subunits, but also to incorporate prior experimental knowledge to locate the MT and γ-TuRC binding sites within the complex, establishing the overall structure of the branching MT organizing center. In addition, we have extended our modeling across diverse eukaryotic genomes and structurally identified augmin homologues, an identification that was not possible previously based on the sequence alone. We demonstrate that the predicted structure of the augmin complex is broadly conserved across four out of five eukaryotic supergroups, suggesting that augmin originated prior to the last eukaryotic common ancestor and that the complex's function may remain broadly conserved.

As we were preparing this manuscript, two additional models of augmin based on cryo-EM density and AlphaFold2-Multimer structural predictions was published: human augmin via cryo-EM[38] and *X. laevis* T-III via cryo-EM supplemented with full-length augmin via negative stain EM[39]. All three models are in general agreement, with overall superposition RMSDs ranging from 2.3 to 6.9 Å, (Supplementary Fig. 7). This agreement is especially important since all these models depend on the interpretation of moderate-resolution cryo-EM density using the cutting-edge but still relatively untested AlphaFold2-Multimer modeling program. Apart from the movement of the poorly resolved second leg of T-III, the largest difference between the models lies in the relative hinging and rotation of T-II. The overall conformation of human T-II is similar to our structure of *X. laevis* T-II, except for a different global rotation of T-II relative to T-III, resulting in a 40 Å displacement at the top of the T-II arch (Supplementary Fig. 7, central inset) and 64 Å in the CH domains (Supplementary Fig. 7, top right inset). This difference could reflect either intrinsic flexibility of this region of augmin or species-specific variability. Our model of *X. laevis* T-II also differed from the two T-II conformers solved by negative stain. The position of our T-II CH domains occupies a location intermediate between the "open" and "closed" conformations described by negative stain (Supplementary Fig. 7, bottom right inset), which could be explained by our T-II map representing a superimposed average between the two conformers. More surprisingly, though, the open and closed conformers of *X. laevis* via negative stain were both were modeled with a opposite-handed kink at the top of the T-II arch than either our model or the human T-II model (Supplementary Fig. 7, central inset). This difference could point to yet another mode of flexibility of T-II or, alternatively, be a result of the differing map resolutions or other technical distinctions between cryogenic and negative stain EM.

In addition to being one of the greatest points of divergence between the published structures, the N-terminus of T-II has also suggested intriguing new hypotheses about augmin's origins and mechanism of action. Two T-II subunits, Haus6 and Haus7, both displayed a classic CH fold at their N-termini, which had not been previously predicted from the protein sequence. The CH fold was first characterized as an actin binder, found in the calcium-dependent myosin regulator calponin. However, a divergent subfamily of CH domains has shifted their affinity from actin to MTs, with the classic example being the MT plus-end binder EB1[40]. In addition to their similarity to all proteins containing tubulin-binding CH domains, Haus6 and Haus7 displayed particularly close homology to members of the so-called NN-CH family (Supplementary Table 2)[41], a diverse group of MT binding multi-subunit complexes. This group includes the NDC80-complex subunits Ndc80 and Nuf2, which are homologs of Haus6 and Haus7. The NDC80 complex captures MTs near the kinetochore, which is also the area of highest augmin activity[42,43]. Thus, it seems plausible that NDC80 and T-II might have diverged from a common ancestral complex deep in the evolutionary past.

Not only does our modeling predict that augmin contains two CH domains, which could potentially serve as additional MT binding sites, but further analysis of our structure yields new insight into which regions of augmin might interact with MTs and how this interaction might support augmin's function. While the CH domain of Haus7 is poorly conserved on the sequence level and is missing altogether in some species including *D. melanogaster*, the CH domain of Haus6 is one of the best-conserved regions of the augmin complex. Examining the predicted structure of Haus6's CH domain more closely, it becomes clear that one face of the domain presents a highly conserved basic surface (Fig. 5a, b). Upon alignment with the CH domain of the Ndc80 subunit, it is apparent that the conserved Haus6 surface overlaps with the MT binding face of Ndc80's CH domain, the primary MT binding site of the NDC80 complex[44]. Although the primary MT binding site of augmin has been shown to reside within the disordered N-terminal 150 residues of Haus8[23], previous results have suggested that T-II harbors a second MT binding site[23] and, taken together, this analysis suggests that the CH domain of Haus6 may be this second MT binding site (Fig. 5c).

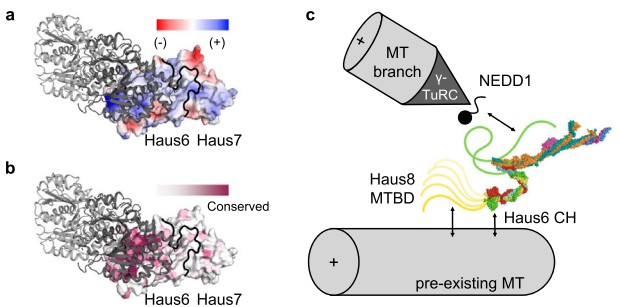

**Fig. 5 | Interaction of augmin complex with the microtubule and other cellular factors. a** Augmin T-II, centered on the calponin homology of Haus6, is displayed as a surface colored by electrostatics (where red is electronegative, and blue is electropositive). A single tubulin dimer, positioned as it would be if bound to the CH domain of NDC80, is displayed in cartoon form (light gray, α-tubulin; dark gray, β-tubulin). **b** Similar to **a**, except that the surface of augmin T-II is colored by amino acid conservation (where purple is conserved and white is variable). **c** Overview of updated model of the MT branch site, including augmin molecular model. The NEDD1 γ-TuRC adaptor binding site is predicted to be located within T-III and/or the C-terminus of Haus6 (green line). Two MT binding site are predicted within T-II, one within the disordered MTBD of Haus8 (yellow line), and a second within the CH domain of Haus6.

One of the major questions about the augmin complex is how it is able to establish the angle of MT branches and why, unlike the actin branches promoted by the Arp2/3 complex[45,46], MT branches can adopt a relatively wide range of branch angles. In particular, prior to obtaining the structure of the augmin complex, it was unclear how augmin could use a highly flexible attachment site— namely the unstructured MTBD of Haus8—to orient γ-TuRC relative to the pre-existing MT. The discovery of a conserved, basic CH domain within Haus6, which is furthermore directly adjacent to the Haus8 MTBD, suggests an answer to this question, namely that the Haus6 CH domain serves as an anchor point to stabilize augmin on the MT and allow the resulting branch to be established at a fixed angle. However, due to the intrinsic flexibility of T-II relative to, e.g., Arp2/3, the complex can likely adopt a range of conformations which, on the macroscopic scale, lead to a range of branch angles. One piece of evidence to link the conformation of T-II with branch angle comes from comparison of our *X. laevis* augmin structure with *D. melanogaster* augmin. AlphaFold2-Multimer predicts that T-II of *D. melanogaster* augmin adopts a much wider arch (~85°) than either *H. sapiens* or *X. laevis* (~50°). Intriguingly, the difference between these two angles is similar to the difference between the range of *D. melanogaster* MT branch angles (30–60°) and vertebrate MT branch angles (0–30°)[18,47]. Thus, our structure suggests that the conformation adopted by T-II may be critical to establishing branching geometry and, more generally, begins to explain how branching MT nucleation can maintain spindle polarity.

## Methods

### Protein expression and purification
Expression of the wild-type augmin Haus6[1–430] complex from Sf9 cells (ATCC) was adapted from a previously described protocol[20]. GFP-tagged versions of augmin subunits were generated by subcloning individual augmin subunits into custom pFASTBAC vectors containing either N-terminal TwinStrep-GFP or C-terminal GFP-TwinStrep tags. Bacmids were then generated by transformation into DH10Bac (NEB) *E. coli*, and subsequent screening on XGAL colorimetric LB plates. Bacmids were purified from DH10Bac culture, screened for insertion by polymerase chain reaction, and transfected into Sf9 cells using standard procedures. Tagged viral stocks were substituted for untagged stocks during co-infection of Sf9 cells with individual viruses bearing each of the eight subunits for augmin co-expression.

Purification of augmin complexes and subcomplexes was adapted from a previously described protocol[20]. 1 L infected Sf9 cells (for T-II and T-III subcomplexes) or 2 L cells (for octameric augmin) were harvested at 500x *g* and resuspended in lysis buffer (50 mM Tris pH 7.7, 200 mM NaCl, 5 mM EDTA, 3 mM β-mercaptoethanol, 10% glycerol, 0.05% Tween 20) supplemented with 10 ug/mL DNase I (Roche) and 1 cOmplete protease inhibitor cocktail tablet (Roche). After lysis via Emulsiflex, lysates were clarified at 200,000 *g* and loaded onto either 1 mL StrepTactin Superflow (IBA), for T-III, or 1 mL IgG Sepharose (Cytiva), for T-II and full augmin complex. After 3 h batch binding, resin was washed with 100 column volumes of lysis buffer and eluted either with 2.5 mM d-desthiobiotin (for T-III) or by overnight cleavage with PreScission protease (for T-II and full complex). Eluates were concentrated using 50 kDa molecular weight cut-off (MWCO) concentrators (Amicon) and further purified over a Superose 6 Increase 10/300 pre-equilibrated in CSF-XB (10 mM HEPES pH 7.7, 100 mM KCl, 5 mM EGTA, 2 mM MgCl₂, 1 mM DTT). Peak fractions were analyzed for purity by SDS-PAGE, pooled, and flash frozen for −80 °C storage.

*X. laevis* γ-TuRC was isolated from *X. laevis* egg extract adapting a recently described purification strategy[24]. Here we utilized magnetic beads (cat. #G7287, Promega, Madison, WI) containing a pre-bound Halo-tagged CM1-containing peptide, Halo-γ-TuNA, to affinity purify native γ-TuRC from egg extract. γ-TuRC was eluted from magnetic beads via protease cleavage using a PreScission protease in a single fraction. The γ-TuRC elution was then concentrated using a 100 kDa MWCO Amicon 4 mL spin concentrator, and further purified with a 10–50% (w/w) sucrose gradient spun in a TLS55 rotor with a Beckman Coulter Optima MAX-XP ultracentrifuge at 200,000 g, 2 °C for 3 h. The sucrose gradient was fractionated, and each fraction was analyzed using western blot and negative-stain EM to determine the peak, which contained 100−150 nM purified γ-TuRC. Purified γ-TuRC was stored in CSF-XB buffer (100 mM KCl, 10 mM K-HEPES, 1 mM MgCl₂, 0.1 mM CaCl₂, 5 mM EGTA, pH 7.7) and ~30% w/w sucrose. Use of *X. laevis* for this study received ethical approval from the Princeton University Institutional Animal Care and Use Committee under protocol 1941.

### Tubulin labeling and polymerization of GMPCPP-stabilized MTs
Bovine brain tubulin was labeled adapting prior methods[48]. Using Alexa568-NHS ester (Invitrogen, A20003) yielded 36–40% labeling efficiency. Single cycled GMPCPP-stabilized MTs were made adapting a previously described protocol[17]. Briefly, 12 μM unlabeled bovine tubulin supplemented with 1 μM Alexa-568 tubulin and 1 μM biotin tubulin was polymerized in BRB80 buffer (80 mM PIPES, 1 mM EGTA, 1 mM MgCl₂) in the presence of 1 mM GMPCPP for 1 h at 37 °C. After 1 h, the MT seed mixture was centrifuged at 13,000 x *g* for 15 min. The supernatant was removed, and the pellet was resuspended in warm BRB80 buffer supplemented with 1 mM GMPCPP.

### Preparation of PEG-functionalized coverslips
22 mm × 22 mm cover glasses (Carl Zeiss, 474030-9020-000) were silanized and reacted with polyethylene glycol (PEG) as previously described[49], except that hydroxyl-PEG-3000-amine and biotin-PEG-3000-amine were used. Glass slides were passivated using poly(L-lysine)-PEG. Flow chambers for TIRF microscopy were prepared using parafilm and gentle heating to seal coverslips to the glass slides.

### Stabilized MT attachment to PEG-functionalized coverslips
Flow chambers were incubated with 5% Pluronic F-127 in water (Invitrogen, P6866) for 5 min at room temperature and then washed with assay buffer (BRB80, 5 mM β-mercaptoethanol, 0.075% (w/v) methylcellulose, 1% (w/v) glucose, 0.02% (v/v) Brij-35 (Thermo Scientific, 20150)) supplemented with 50 μg/mL κ-casein. Flow chambers were then incubated with an assay buffer containing 50 μg/mL NeutrAvidin (Invitrogen, A2666) for 2 min on a metal block on ice and subsequently washed with BRB80. Next, flow chambers were incubated for 5 min at

room temperature with Alexa-568 labeled, biotinylated GMPCPP-stabilized MTs diluted 1:2000 in BRB80. Unbound MTs were removed by additional BRB80 washes.

## TIRF microscopy and image analysis

For in vitro branching MT nucleation reactions, a mixture of augmin-Haus6[1–430] (50 nM) and γ-TuRC (10 nM), was first incubated for 5 min on ice, and then flowed into the imaging chamber which contained biotin labeled Alexa-568 GMPCPP-stabilized MTs. Components were allowed to incubate within the imaging chamber for 5 min at room temperature to allow for localization of all branching components to stabilized MTs prior to initiating the reaction. After incubation, unbound proteins were removed by washing the reaction chamber with BRB80 (80 mM PIPES, 1 mM MgCl$_2$, 1 mM EGTA, pH 6.8 with KOH).

TIRF microscopy was performed with a Nikon Ti-E microscope using a 100 × 1.49 NA objective. Andor Zyla sCMOS camera was used for acquisition, with a field of view of 165.1 × 139.3 μm, multi-color images were acquired using NIS-Elements software (Nikon). All adjustable imaging parameters (exposure time, laser intensity, and TIRF angle) were kept the same within experiments. During data acquisition of in vitro branching MT nucleation reactions, the TIRF objective was warmed to 33 °C using an objective heater (Bioptechs, 150819–13), and data was collected using time-lapse imaging, multi-color images collected every 2 sec. ImageJ version 2.3.0/1.53f[50] was used for image processing and data analysis.

## EM data collection

Negative-stained EM samples were prepared by diluting purified augmin to 150 nM in CSF-XB and pipetting 3 μl onto glow-discharged (15 mA, 25-30 secs) carbon film, 400 mesh Cu grids (Electron Microscopy Sciences), staining with 0.75% uranyl acetate solution. Negative-stain EM data was collected at 94,000x magnification (1.56 Å/pixel) with single-tilt using a Talos F200X Transmission Electron Microscope equipped with a 4k x 4k Ceta 16 M CMOS camera.

Cryo-EM grids were prepared similarly using undiluted, purified augmin. 0.05% NP-40 was added to augmin prior to applying to grids. Here, 3 μl of sample was applied to glow-discharged (10 mA, 8 sec) Quantifoil holey carbon R 1.2/1.3 400 mesh grids coated with a homemade thin carbon film (~5 nm thickness) using Leica EM ACE600 High Vacuum Sputter Coater. The grids were flash frozen in liquid ethane using a FEI Vitrobot Mark IV (Thermo Scientific) plunge freezer, using a blot force of 0 and with a 4.5 sec blot time. Cryo-EM data were collected using the Titan Krios microscopes at either Washington University in St. Louis (WUSTL) or Case Western Reserve University (CWRU). The data collection parameters are listed in Supplementary Table 1.

## EM data processing

Data processing of negative-stain EM data was done using Relion 3.0.6[51]. Here, raw micrographs were used to manually pick particles for alignments and averaging. ~10,000 total particles were manually picked for each tagged complex, followed by particle extraction and 3–10 rounds of 2D class averaging.

Data processing of cryo-EM data (diagrammed in Fig. 2a) was done using CryoSPARC v3.3.2 + 220824[52]. Raw micrograph movies were motion corrected and CTF-corrected using CryoSPARC Live's motion correction algorithm and Patch CTF, respectively. Augmin templates were generated from negative stain class averages of full length augmin and augmin T-III and independently used to pick particles, to account for model bias from template picking. After particle extraction and 2D classification, both sets of classes displayed the characteristic "h" shape of the full augmin complex. Particles from the best 3 classes (amounting to 9,385 particles) were used to generate an ab initio model. Evenly spaced templates were generated by CryoSPARC and then a second round of template picking was used to pick

~2,000,000 particles. These particles were sorted extensively by 2D classification, and the best 114,100 were used for one round of homogenous 3D refinement with dynamic masking. The resulting sharpened map was used for all subsequent analysis.

## AlphaFold2 structural prediction and model docking

Canonical isoforms and sequences of *X. laevis* augmin subunits were input either singly into AlphaFold2 2.1[26] or in multimeric groups, using the --*multimer* option. Each of the resulting five independent T-II and T-III models were manually placed at three separate locations and rigid-body docked into the cryo-EM map and its inverse-hand equivalent using ChimeraX fitmap[53]. Cross-correlation scores from fitmap were used to determine the correct map hand and the best-fitting model out of the original five. Structural alignments were performed using Cα superposition in PyMOL. Surface conservation of T-II was calculated using the ConSurf server[54]. Structure figures were generated in either ChimeraX[53] or PyMOL (Version 1.8, Schrödinger, LLC).

Orthologues of augmin subunits were identified through a combination of literature review, BLAST search using the domain enhanced lookup time accelerated-BLAST algorithm[55], and HMMR hidden Markov model search[56]. Orthologues that had not been experimentally verified were validated for sequence completeness by alignment to their closest 10 homologs via BLAST search in the full UniProt database and verified as non-spurious augmin orthologs by targeted BLAST search in the *X. laevis* or *A. thaliana* genome to ensure that the expected augmin subunit was the top hit. Structure-based homology search was performed using the DALI server[30], searching either the full PDB experimentally determined database or the full *H. sapiens* AlphaFold2 predicted genome[31].

## Reporting summary

Further information on research design is available in the Nature Portfolio Reporting Summary linked to this article.

## Data availability

The reconstructed electron density map can be obtained from the EMDB using accession code EMD-28981. The rigid-body fit structural model associated with this map can be obtained from the PDB using accession code 8FCK. All other data are available from the corresponding author on request.

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

## Acknowledgements

We would like to thank current and former members of the Petry lab for support and scientific feedback, particularly Collin McManus and Venecia Valdez for critical reading of the manuscript. We thank Michael Rau and James Fitzpatrick at WUCCI and Kunpeng Li at CWRU for microscopy support. We are grateful to Alan Brown, Xiao Fan, Fred Hughson, and Phil Jeffrey for helpful scientific discussions. We would also like to thank the staff of the Princeton Imaging and Analysis Center, particularly John Schreiber, for their technical assistance in electron microscopy, and Matthew Cahn of Princeton Research Computing for his assistance with data processing and AlphaFold2 implementation. This work was supported by National Institutes of Health grants F32GM142149 (SMT), the Helen Hay Whitney Foundation (JK), NIGMS grant 1R01GM138854 (RZ), and NIGMS grant R011R01GM141100-01A1 (SP). The authors acknowledge the use of Princeton's Imaging and Analysis Center (IAC), which is partially supported by the Princeton Center for Complex Materials (PCCM), a National Science Foundation (NSF) Materials Research Science and Engineering Center (MRSEC; DMR-2011750). Molecular graphics and analyses performed with UCSF Chimera, developed by the Resource for Biocomputing, Visualization, and Informatics at the University of California, San Francisco, with support from NIH P41-GM103311.

## Author contributions

B.P.M., R.Z., and S.P. conceived of the project, and D.J.T., R.Z., and S.P. supervised the project. S.T. and B.P.M. prepared protein samples. S.T., B.P.M., W.H., and M.M. prepared cryo-EM grids and collected cryo-EM data. S.T., B.P.M., and W.H. processed cryo-EM data. S.T. and W.H. performed modelling in AlphaFold2 and structure validation. J.K. and M.J.R. collected and processed TIRF data. S.T. and B.P.M. wrote the paper. All authors edited the manuscript.

## Competing interests

The authors declare no competing interests.
