## [Peer Review File · Nature Communications]

REVIEWER COMMENTS

Reviewer #1 (Remarks to the Author):

The activity of the augmin complex in promoting microtubule branching is important for cytoskeleton regulation particularly during cell division. As pointed out by the authors, the lack of structural information about this complex has limited definition of the molecular basis of this important cellular activity. The hetero-octameric augmin complex is a challenging target for structural studies and the authors take an integrated approach, using cryo-EM, judicious use of AlphaFold2 to build a model of the *Xenopus laevis* augmin complex, computational docking and protein engineering. The protein complex they analyze lacks the C-terminus of the Haus6 subunit, but is validated to be functionally active with respect to microtubule branching by the authors.

The manuscript is generally clearly written and provides an overall convincing account of the data. The limitations of some of the approaches are appropriately acknowledged - for example the relatively modest resolution of the cryo-EM density - and the resulting interpretations are suitably measured. The evolutionary analyses arising from the work are intriguing and will pave the way for functional studies to further evaluate augmin activity in cellular contexts. Having said that, parts of the Results text are rather descriptive and hard to follow in places, while several other aspects of the Methods and data analysis would benefit from a more complete account. Versions of software and proper referencing of these are frequently missing and must be included.

The points below outline the ways in which the manuscript can be improved to provide a more complete explanation of the work performed.

1. At the beginning of the Results section, it would be useful to provide a brief description of what is known about the C-terminus of Haus6, which is excluded from the augmin construct used in this study – is it conserved, what size is it, does it have any predicted structural features?

2. A more thorough depiction of the steps towards cryo-EM reconstruction following standards in the field should be provided – this should include an exemplar micrograph and inclusion of more 2D classes than are shown in current Figure 1c. It would also be instructive and appropriate to see examples of particles used to train the machine learning particle picking algorithm, especially given that picking problems and the high background are mentioned in the text.

3. The description of the procedure by which the AlphaFold2 models have been selected and presented should be more completely described. How much variability was there amongst the five AlphaFold2-generated models mentioned in the Methods? What were the criteria by which model(s) were selected for further use? An additional view in Figure 2b of the overall fit should be provided and it should be made clear in the figure legend that what is depicted is the result of manual docking. Annotation of key features in the figure and which are also referred to in the text (bulge in the T-III stalk, main leg, second leg) would also avoid ambiguity and facilitate reader comprehension.

4. The account of how the AlphaFold2 models were fitted into the cryo-EM density using MDFF and how this fitting has been validated is very descriptive and incomplete. To further support and clarify interpretation of the cryo-EM structure, the MDFF experiments, currently outlined in Sup Fig 3 should be presented in a main figure, together with quantitative validation of improvement of the fits to the experimental density. What is the evidence for a single unique solution following flexible fitting, especially given the modest cryo-EM resolution? Do the MDFF results correlate with structural variability evidenced in the AlphaFold2 predictions? These comparisons should be discussed. Further, even at the resolution of the presented cryo-EM structure, discrete helical densities should be visible and match the fitted model. Exemplar zoomed in regions of density/model should be shown as support for the structural analyses.

5. Introduction of a compact, globular GFP at multiple points in the augmin complex is an elegant and convincing way to further validate the structural model(s). However, the presentation of these data in Figure 3 is not clear, with the corresponding main text adding to the confusion. Referring to Figure 3a on p7 in the Integrated structural model section is also confusing and disrupts the narrative flow. Figure 3 can be further improved by:

i) explicitly defining the construct nomenclature - 1-4 C or N is not referenced in the results text or figure legends at all; ii) inclusion of no GFP control construct as a point of comparison; iii) inclusion of a scale bar; iv) a clearer explanation of why some samples show protein density to different extents; v) a clearer explanation of the models depicted above the EM 2D classes, including why some parts of the complex are only depicted in pale coloring.

6. MDFF is not mentioned in the Discussion and it should be i) added to the list of approaches used in paragraph 1 and ii) mentioned more in the context of discussion of the differences between AlphaFold2 predictions and the cryo-EM density in paragraph 2.

7. The Discussion makes reference to one very recent publication describing human augmin (Gabel et al), but there is another *Xenopus laevis* augmin structure publication (Zupa et al) which should also be discussed and compared to the current work.

Minor points

- 1st paragraph of Results: typo “thus stimulated branching MT nucleation..”
- throughout: AlphaFold should be AlphaFold2 and CryoSparc should be CryoSPARC
- the version of ImageJ and appropriate reference should be provided in the Methods
- the version of CryoSPARC and appropriate reference should be provided in the Methods
- the version of RELION should be provided in the Methods
- the version of Topaz should be provided in the Methods
- the reference for Chimera fit-in-map should be provided in the Methods
- the reference for Pymol should be provided in the Methods
- Fig 1c – scale bar missing

Reviewer #2 (Remarks to the Author):

The authors present here a model of the augmin complex based on a low-resolution cryo-EM structure, negative stain EM of complexes with bulky tags and docking of AlphaFold-predicted protein structures. The manuscript is well written, and the model of the structure agrees largely with a previously published structure from the Barford lab and a notable difference between the two models is discussed.

This reviewer was asked to specifically comment on the TIRF microscopy data (Fig. 1, Fig. S1):

1. The authors use a purified HAUS complex lacking part of HAUS6 (for solubility reasons) and claim in Fig. S1 that augmin with this truncation together with gammaTuRC can support branched microtubule nucleation. The authors may want to clarify if they want to claim that the truncated part is not needed for augmin's function or what can be learnt for the function of the truncated part from their experiments.
2. The authors seem to contradict previous work from the Petry lab where the conclusion was made that branching microtubule nucleation critically requires TPX2 in addition to augmin and

gammaTuRC, using *Xenopus* proteins (Alfaro-Aco et al., *Elife*, 2020). Recently, an *in vitro* reconstitution study using human proteins suggested that TPX2 was not required for branching (Zhang et al., *JCB*, 2022), leaving the possibility that the branching mechanism differs between human and *Xenopus*. For clarity, the authors should state whether their current reconstitution using *Xenopus* proteins revises some of the previous conclusions drawn in the Alfaro-Aco paper and instead confirms what has been concluded regarding the role of TPX2 in the Zhang et al paper.

3. Fig. 1b shows cropped versions of the images presented in Fig. S1b bottom. That should at least be openly stated. Is this the only experiment that the authors can show as an example for branched microtubule nucleation? In both legends, the scale bar is stated to indicate a distance in 'micromolar'. Please correct.

4. In the images as presented, one can appreciate that addition of gTuRC alone induces microtubule nucleation in solution (not originating from pre-existing microtubules). Branched microtubule nucleation could be shown more convincingly if conditions are used that cause less spontaneous microtubule nucleation in solution (because spontaneously nucleated microtubules in the vicinity of other microtubules may appear as branched microtubules). Why do microtubules only branch from the pre-immobilized GMPCPP microtubule and not also from branched microtubules as observed in a previous reconstitution (Zhang et al, *JCB*, 2022)? Is branching nucleotide or fluorophore-dependent?

5. In contrast to the 'gTuRC only' condition, augmin and gTuRC together induce the appearance of numerous tubulin "particles" on the surface that do however not appear to contain augmin - at least not visibly, given the contrast and brightness adjustments of the images as presented. How can the appearance of these tubulin particles be explained?

6. An important control experiment is missing: the addition of augmin alone (without gTuRC). The authors conclude that unlabelled gTuRC may be recruited by augmin, because they observe signs of microtubule branching from pre-formed microtubules when both gTuRC and augmin are added. But one cannot exclude that under the conditions used augmin alone is responsible for the branching effect, given the data as presented.

7. The authors claim that the expected acute branching angles are observed. In the presented image of microtubule branching shown in Fig. S1 this does however not seem to be the case. Microtubules appear to branch off at various angles. More example images need to be presented and analyzed if the authors want to make statements about the branching angles, ideally under conditions with fewer spontaneously nucleated microtubules.

8. Given that the authors have previously published experiments with fluorescently labelled gTuRC they could directly show the recruitment of gTuRC by the truncated augmin complex, instead of inferring it indirectly.

9. The temperature at which the branching experiments are performed should be stated, because it is an important parameter influencing microtubule nucleation. The tubulin concentration is relatively high, and one may expect that microtubules nucleate spontaneously at this concentration unless the temperature is fairly low.

10. The fluorescence image showing the augmin channel is presented using contrast and brightness settings that appear to suppress any background signal and strongly enhance the contrast. A gentler way of brightness and contrast adjustment would probably give a more realistic representation of the situation.

Other comments:

11. It would be useful to indicate in at least one of the figures where in the augmin structure the missing HAUS6 piece is located and indicate schematically how big it may be.

12. The authors discuss a difference between the recently published model for the structure of the augmin complex (Gabel et al., Nat. Comms., 2022) and cite Fig. S4. However, no comparison between the two structures is shown in this figure. It would be useful to illustrate the difference in the position of the calponin homology domains in the two models in a figure more directly.

13. The discussion about augmin in different species with the evolutionary implications is interesting. But a discussion about what we can learn from the structure of augmin, as presented here, for augmin's function is largely missing.

Reviewer #3 (Remarks to the Author):

Travis et al present an elegant study of the augmin complex, focusing on its structure, structure/function relationship and the structural conservation across eukaryotes. It is distinct from the two previously submitted manuscripts focusing on augmin structure through the particularly extended evolutionary biology component - not this reviewer's area of expertise. The demonstrated structural insight sheds a convincing light on the function of the complex.

Suggestions for revision:

1. In the abstract the authors claim that branching microtubule nucleation "exponentially amplifies microtubules" - this implies a defined quantitative relationship, but has this really been quantified to demonstrate that it is indeed "exponential"? If not strictly supported by evidence, I suggest changing the wording to a less specific term

2. Also in the abstract: "branching has been hindered by a lack of structural information" - shouldn't it be "the"?

3. Abstract and beyond: "allowed" implies that an object has authority, perhaps using "enabled" is more appropriate

4. The introduction seems under-referenced in places:

"Critically, spindle MTs have to be continuously generated, as any given MT turns over within seconds, yet the spindle framework must often persist for up to an hour. Due to the spindle's central importance for viability, multiple partially redundant pathways are used to generate spindle MTs."

Each statement should be referenced and some examples of these multiple (redundant) pathways might also be helpful for a more general reader.

5. Also, terms such as "spindle body", "spindle assembly checkpoint" may need some brief introduction or definition

6. "with the bottom of the "h" sitting on the MT" - sure not "an MT"?

7. "In addition, lack of any identified structural homologs" - perhaps "the lack"?

8. Results: "Expanding upon our results with *D. melanogaster* augmin, we next sought to answer another major unknown question about the augmin complex" - "unknown question" sounds a bit pleonastic

9. Discussion: "This model has allowed us not only to identify the positions and orientations of all eight subunits, but also to incorporate prior experimental knowledge to locate the MT and γ -TuRC binding sites within the complex, establishing for the first time the overall structure of the branching MT organizing center" - better to avoid the statement of priority here, even if true (but probably not)

10. "Haus6 and Haus7 displayed, unexpectedly, a classic calponin homology fold at their N-termini." - why "unexpectedly"?

11. "This second MT binding mode may help solve the mystery of how a disordered attachment via the unstructured N-terminus of Haus8 is capable of orienting γ -TuRC stably relative to the pre-existing MT, and more generally how branching MT nucleation is capable of maintaining spindle polarity" - would not call it a second binding mode in this context, suggest rephrasing to something like:

"The second, firm/fixed anchor point may explain how branching..." - would be nice to give this concept a bit more emphasis in the figure, if possible, maybe by including some more arrows and labels in Fig 5a?

12. Have not found any reference to Fig 5c in the text - what is illustrated should be worth discussing, I guess?

13. Not sure if Materials and Methods provide enough information to replicate the study. For example, haven't found how tubulin was labelled - procedure or the commercial source should be provided.

14. "Schrödinger" not "Schrodinger"

15. Fig 1b - the most important bits are strictly invisible when printed. On the screen it is better, but the panels are too small and contrast is not great, perhaps close-ups would be good as well as larger overviews.

16. Fig 2 legend: "Final model of T-II and T-III into moderate resolution cryo-EM map" - missing "fitted"?

17. Fig 3 legend: top/bottom panels maybe better than "above/below"

18. Fig 5a legend: "The NEDD1 γ -TuRC adaptor binding site is predicted to be located with T-III (pink)" - should be "within"?

19. Fig S1: close-up views would be good if not included in the main figure

We are very grateful to the reviewers for giving such helpful feedback and recommendations. Your comments have substantially aided us in improving the structure presented, especially in terms of the model fitting and map interpretation, as well as given us helpful guidance in how to best present and discuss our results in both the text and the figures. In particular, by re-examining the multiple models output by AlphaFold2-Multimer, we were able to find predicted conformers of both augmin tetramers that much better fit our map. This provided us an alternative approach to MDFF, allowing us to avoid the danger of overfitting. In addition, by revisiting the negative stain data, we were able to resolve the N-termini of two additional subunits, Haus3 and Haus5, clarifying the position of a previously poorly-modeled region of the augmin structure. We have substantially revised both the text and the figures to incorporate the suggestions of the reviewers, which we feel has resulted in a much clearer manuscript with additional analysis.

Specifically, we have made the following changes in response to reviewer feedback:

Reviewer #1:

1. At the beginning of the Results section, it would be useful to provide a brief description of what is known about the C-terminus of Haus6, which is excluded from the augmin construct used in this study – is it conserved, what size is it, does it have any predicted structural features?

We thank the reviewer for this comment and have added a short description of the Haus6 C-terminal region to the first paragraph of the results section. In addition, we have added a cartoon depiction of the region to our model figures (Figure 2, center, and Figure S4a) in order to aid the reader in understanding which parts of the augmin complex were visualized in our reconstruction, and which were unmodeled.

2. A more thorough depiction of the steps towards cryo-EM reconstruction following standards in the field should be provided – this should include an exemplar micrograph and inclusion of more 2D classes than are shown in current Figure 1c. It would also be instructive and appropriate to see examples of particles used to train the machine learning particle picking algorithm, especially given that picking problems and the high background are mentioned in the text.

This is an excellent point, which we addressed in the following way. We have substantially added to supplementary figure S2 to depict the steps towards cryo-EM reconstruction, including an exemplar micrograph with particle picks and 2D class averages. We have also included additional representative 2D class averages in the main text in Figure 1c.

In addressing point #4 below, regarding the resolution of the final map, we revisited the particle picking strategy to determine whether we could improve the effective map quality to better match the resolution calculated by CryoSPARC. Unfortunately, as discussed below, the effective resolution of the map remains lower than CryoSPARC suggests. However, as a side effect of this analysis, we were able to reduce bias in particle orientation by replacing the machine learning particle picking step with a second template-picking step, namely generating evenly-spaced templates from an ab initio 3D model. We have revised Figure S2 and the

second paragraph of the Results, as well as the Methods section on cryoEM data processing, to reflect this change.

3. The description of the procedure by which the AlphaFold2 models have been selected and presented should be more completely described. How much variability was there amongst the five AlphaFold2-generated models mentioned in the Methods? What were the criteria by which model(s) were selected for further use? An additional view in Figure 2b of the overall fit should be provided and it should be made clear in the figure legend that what is depicted is the result of manual docking. Annotation of key features in the figure and which are also referred to in the text (bulge in the T-III stalk, main leg, second leg) would also avoid ambiguity and facilitate reader comprehension.

We appreciate this reviewer's comments. We have added a supplementary figure (Figure S3) to provide more information about the AlphaFold2 modeling and model error, selection of model, and rigid-body model docking. We have also revised the main text (paragraph 3 of the "Integrated Structural Model" section) to further discuss how variability between AlphaFold2 models was used to improve agreement between structure and map. We have modified Figure 2 to include a second view of the complex. In Figure 2, we have annotated the main features of the augmin complex discussed in the text. Finally, we have revised the figure legend to clarify that we are depicting the results of manual docking followed by rigid body fitting.

4. The account of how the AlphaFold2 models were fitted into the cryo-EM density using MDFF and how this fitting has been validated is very descriptive and incomplete. To further support and clarify interpretation of the cryo-EM structure, the MDFF experiments, currently outlined in Sup Fig 3 should be presented in a main figure, together with quantitative validation of improvement of the fits to the experimental density. What is the evidence for a single unique solution following flexible fitting, especially given the modest cryo-EM resolution? Do the MDFF results correlate with structural variability evidenced in the AlphaFold2 predictions? These comparisons should be discussed. Further, even at the resolution of the presented cryo-EM structure, discrete helical densities should be visible and match the fitted model. Exemplar zoomed in regions of density/model should be shown as support for the structural analyses.

We thank the reviewer for raising these points and agree that, at the resolution estimated by CryoSPARC, discrete helical densities should be visible and that this helical density should be sufficient to guide MDFF experiments. We revisited the data processing, particularly the particle picking steps, and, as discussed in response to point #1, determined that an alternate particle picking strategy modestly improved the resolution estimated by CryoSPARC from 8.4 Å to 6.9 Å, as well as somewhat decreasing particle anisotropy. However, we were still unable to visualize clear helical density, even in the best resolved portions of the map near the base of T-III.

This discrepancy was initially confusing to us, and, after extensive discussion both amongst the authors and with colleagues expert in cryoEM but not involved with the project, we came to the conclusion that CryoSPARC's estimation of resolution was not consistent with the map

quality we were observing. Colleagues anecdotally reported that CryoSPARC often overestimates the resolution of maps in the moderate resolution range, and we suspect that is what is occurring here. This systematic error on the part of CryoSPARC, combined with the anisotropy of our data imposed by preferred orientation (as visualized in supplementary figure S2c), likely means that the true resolution of our map is somewhat lower than what CryoSPARC reports.

To address this issue, we added a section to the end of the second paragraph of the Results section highlighting this resolution discrepancy for the reader as well as our theories as to the possible cause. In addition, we have added inset figures to Figure 2, highlighting both regions of good agreement and poor agreement between model and map, to aid readers in assessing map quality and model fit. Finally, as discussed below in response to point #6, we decided to remove the MDFF experiments from the manuscript, as MDFF no longer seemed appropriate at this lower effective resolution. We hope these changes will address the reviewer's concerns.

5. Introduction of a compact, globular GFP at multiple points in the augmin complex is an elegant and convincing way to further validate the structural model(s). However, the presentation of these data in Figure 3 is not clear, with the corresponding main text adding to the confusion. Referring to Figure 3a on p7 in the Integrated structural model section is also confusing and disrupts the narrative flow. Figure 3 can be further improved by:

i) explicitly defining the construct nomenclature - 1-4 C or N is not referenced in the results text or figure legends at all; ii) inclusion of no GFP control construct as a point of comparison; iii) inclusion of a scale bar; iv) a clearer explanation of why some samples show protein density to different extents; v) a clearer explanation of the models depicted above the EM 2D classes, including why some parts of the complex are only depicted in pale coloring.

We thank the reviewer for their helpful suggestions for improving this figure and its discussion:

i. We have revised both the figure and the legends to improve clarity and consistency of the labeling nomenclature.

ii. We have included a no-GFP control construct of T-III in Figure 3a. For T-II, as a GFP tag on Haus2 is required for purification and is found on both our cryo-EM constructs and all four tagged constructs presented in Figure 3c, we have included a T-II control singly tagged at the C-terminus of Haus2 for purposes of comparison.

iii. We have added a scale bar for ease of reader interpretation.

iv. We have revisited the data processing for two of the T-III constructs, doubly-labeled with Haus1-C-GFP and either Haus3-N-GFP or Haus5-N-GFP. In so doing, we have found additional 2D classes displaying density consistent with the missing GFP density at the N-termini of Haus3 and Haus5. We have revised the text to reflect this new analysis. We have

also revised the text to better discuss why density is missing for the C-terminus of T-II and the N-terminus of Haus8. We hope that these revisions have addressed the reviewer's concerns.

v. We have reworked how the models above the EM classes are depicted to improve readability and in particular have added explanations for why some regions are depicted in pale coloring.

We have also reworked the presentation of data in the "Integrated Structural Model" section to improve manuscript flow and better match the progression of the figures. Specifically, we have added a reference to the cryo data presented in Figure 1 to the "Integrated Structural Model" section, and returned to our discussion of flexibility as visible in the negative stain data in the "Model Validation through GFP-tagging" section.

6. MDFF is not mentioned in the Discussion and it should be i) added to the list of approaches used in paragraph 1 and ii) mentioned more in the context of discussion of the differences between AlphaFold2 predictions and the cryo-EM density in paragraph 2.

We thank the reviewer for bringing this to our attention. As discussed in response to point #4, we decided that MDFF was not appropriate at the effective resolution of our map, and thus decided to remove the MDFF experiments from our text.

7. The Discussion makes reference to one very recent publication describing human augmin (Gabel et al), but there is another Xenopus laevis augmin structure publication (Zupa et al) which should also be discussed and compared to the current work.

We thank the reviewer for bringing this to our attention and agree that this second publication, which we only learned of after submitting our manuscript, should absolutely be discussed and compared with the current work. We have expanded the second paragraph of the Discussion to encompass both recent publications and summarize the new state-of-the-field with regards to the structure of augmin. In response to Reviewer #2's suggestion, we have also added a supplementary figure (Figure S7) to aid the reader in comparing these structures.

Minor points

- 1st paragraph of Results: typo "thus stimulated branching MT nucleation.."

We have fixed this typo.

- throughout: AlphaFold should be AlphaFold2 and CryoSparc should be CryoSPARC

We have fixed these typos.

- the version of ImageJ and appropriate reference should be provided in the Methods

We have added the version information and reference for ImageJ in the Methods section.

- *the version of CryoSPARC and appropriate reference should be provided in the Methods*

We have added the version information and reference for CryoSPARC to the Methods.

- *the version of RELION should be provided in the Methods*

We have added the version information for Relion to the Methods.

- *the version of Topaz should be provided in the Methods*

As discussed in response to point #1, we have revised our processing workflow and are no longer using Topaz. Thus, we have removed references to Topaz from the Methods.

- *the reference for Chimera fit-in-map should be provided in the Methods*

We have added the reference for Chimera *fitmap* to the Methods.

- *the reference for Pymol should be provided in the Methods*

We have added the reference for PyMOL to the Methods.

- *Fig 1c – scale bar missing*

We have added a scale bar to Figure 1c for ease of reader interpretation.

Reviewer #2:

1. The authors use a purified HAUS complex lacking part of HAUS6 (for solubility reasons) and claim in Fig. S1 that augmin with this truncation together with gammaTuRC can support branched microtubule nucleation. The authors may want to clarify if they want to claim that the truncated part is not needed for augmin's function or what can be learnt for the function of the truncated part from their experiments.

We thank the reviewer for their feedback. We have added a sentence to the first paragraph of the results section to clarify what we hope to learn from the truncated complex, namely that, because the C-terminus of Haus6 appears to be functionally redundant with T-III, we expected to see wildtype-like activity for the truncated complex, verifying that the complex used for structural determination was biochemically functional.

2. The authors seem to contradict previous work from the Petry lab where the conclusion was made that branching microtubule nucleation critically requires TPX2 in addition to augmin and gammaTuRC, using Xenopus proteins (Alfaro-Aco et al., Elife, 2020). Recently, an in vitro reconstitution study using human proteins suggested that TPX2 was not required for branching (Zhang et al., JCB, 2022), leaving the possibility that the branching mechanism differs between

human and Xenopus. For clarity, the authors should state whether their current reconstitution using Xenopus proteins revises some of the previous conclusions drawn in the Alfaro-Aco paper and instead confirms what has been concluded regarding the role of TPX2 in the Zhang et al paper.

We thank the reviewer for pointing this out. We have expanded the first paragraph of the Results section, which describes the results of the biochemical reconstitution, to explain the context of these results as compared to our previous work (Alfaro-Aco et al 2020 eLife) as well as the more recent Zhang et al study.

3. Fig. 1b shows cropped versions of the images presented in Fig. S1b bottom. That should at least be openly stated. Is this the only experiment that the authors can show as an example for branched microtubule nucleation? In both legends, the scale bar is stated to indicate a distance in 'micro-molar'. Please correct.

We thank the reviewer for catching this. We have modified the legend of Figure 1 to explicitly state that the images have been cropped for clarity and corrected the scale bar to indicate units of microns and not micromolar. We have repeated the experiments depicted in Figures 1 and S1 several times, but we still believe that the original images, which we have retained in Figures 1 and S1, best depict the data. However, as noted in response to point #10, we have adjusted the brightness and contrast, which should help the reader visualize the data more clearly.

4. In the images as presented, one can appreciate that addition of gTuRC alone induces microtubule nucleation in solution (not originating from pre-existing microtubules). Branched microtubule nucleation could be shown more convincingly if conditions are used that cause less spontaneous microtubule nucleation in solution (because spontaneously nucleated microtubules in the vicinity of other microtubules may appear as branched microtubules). Why do microtubules only branch from the pre-immobilized GMPCPP microtubule and not also from branched microtubules as observed in a previous reconstitution (Zhang et al, JCB, 2022)? Is branching nucleotide or fluorophore-dependent?

We thank the reviewer for their feedback. We have added several details in the first paragraph of the Results section to clarify our experimental set-up. Many of the specifics of our assay differ from the reconstitution by Zhang et al. In our reconstitution set-up, instead of adding soluble tubulin and branching factors at once, we perform a two step reaction: first, we bind the branching factors to the GMPCPP seed, and only after washing out unbound branching factors do we add soluble tubulin. We have found that this two-step reaction is necessary to reduce non-specific nucleation off the MT seed, but it has the side effect of also preventing the second round of MT branching observed by Zhang et al.

As to the concentration of tubulin, we chose to work at a relatively high concentration, 20 μ M, so that we could observe a maximal number of branches. With lower tubulin concentrations, it is difficult to unambiguously quantify MT polarity due to the low number of MT. It is worth

noting that 20 μM tubulin is also the concentration used in the 2022 reconstitution by Zhang et al. To decrease background nucleation, we might instead drop the concentration of $\gamma\text{-TuRC}$ used in the reconstitution experiments. However, because we are working with truncated augmin, lacking the secondary $\gamma\text{-TuRC}$ binding site found within the C-terminus of Haus6, we observe reduced affinity of augmin for $\gamma\text{-TuRC}$ compared to the full-length complex (unpublished observations), and thus using a lower $\gamma\text{-TuRC}$ concentration is not feasible. We have edited the first paragraph of the Results section to explicitly make this point.

In terms of whether branching is nucleotide- or fluorophore-dependent, we have previously observed that the branching factor TPX2 is preferentially recruited to GMPCPP seeds over new MT branches (Alfaro Aco et al 2020 eLife). However, augmin demonstrates no nucleotide preference, so this is unlikely to be a reason why we only observe daughter branches, and not the granddaughter branches seen in the Zhang et al 2022 paper. In addition, we have performed our reconstitution experiments with the tubulin fluorophores inverted (ie Cy5-tubulin seeds and Alexa-568 soluble tubulin) and observed the same results, so we believe the fluorophore has no effect on branching. We obtain the best results with Alexa-568 labeled seeds and Cy5-labeled soluble tubulin because the contrast of Alexa-568 is higher than that of Cy5, making it easier to observe the seed signal within the bright halo of soluble tubulin recruited by augmin that surrounds the seed.

5. In contrast to the 'gTuRC only' condition, augmin and gTuRC together induce the appearance of numerous tubulin "particles" on the surface that do however not appear to contain augmin - at least not visibly, given the contrast and brightness adjustments of the images as presented. How can the appearance of these tubulin particles be explained?

The reviewer raises a good point. We suspect that these tubulin particles are an artifact of the contrast and brightness adjustments of the images. As we mentioned in the text, our data suggest that augmin is capable of recruiting soluble tubulin to the GMPCPP seeds, since we see the MT seeds coated with tubulin under this condition. The seeds coated with tubulin are much brighter than the MT branches formed, so the brightness and contrast was adjusted to visualize the branches. Even in the buffer only condition, one can appreciate some tubulin particles on the background of the glass. These are not as apparent in the $\gamma\text{-TuRC}$ only condition because $\gamma\text{-TuRC}$ spontaneously nucleates more MTs, which appear brighter against the background. However, this is only one possible explanation. Ultimately, because the particles do not contain augmin and are spatially distinct from augmin-nucleated branches, we feel that determining their mechanism of formation and biological relevance is beyond the scope of this paper, which focused exclusively on the structure and function of the augmin complex in branching. However, we intend to revisit this result in subsequent work.

6. An important control experiment is missing: the addition of augmin alone (without gTuRC). The authors conclude that unlabelled gTuRC may be recruited by augmin, because they observe signs of microtubule branching from pre-formed microtubules when both gTuRC and augmin are added. But one cannot exclude that under the conditions used augmin alone is responsible for the branching effect, given the data as presented.

We thank the reviewer for pointing this out and agree that, based on the data presented, the reader could conclude that augmin alone can support MT branching. However, multiple groups, including our own, have previously examined whether augmin is itself a MT nucleator and found that the complex has no intrinsic ability to nucleate MTs. We have added a sentence to paragraph 2 of the results section to highlight this fact for the reader, along with the relevant citations. We hope that this addition will address the reviewer's concern.

7. The authors claim that the expected acute branching angles are observed. In the presented image of microtubule branching shown in Fig. S1 this does however not seem to be the case. Microtubules appear to branch off at various angles. More example images need to be presented and analyzed if the authors want to make statements about the branching angles, ideally under conditions with fewer spontaneously nucleated microtubules.

We thank the reviewer for bringing this to our attention and agree that the reconstitution data presented is insufficient to determine the angle of the resulting MT branches. We have thus removed the claim in the first paragraph of the Results section that the angles formed are acute, as we feel that determining branch angle lies outside of the scope of this study.

8. Given that the authors have previously published experiments with fluorescently labelled gTuRC they could directly show the recruitment of gTuRC by the truncated augmin complex, instead of inferring it indirectly.

We thank the reviewer for their suggestion of this experiment, as we have indeed used this set-up before to directly visualize recruitment of γ -TuRC to augmin. However, we have also previously observed that fluorescent labeling of γ -TuRC can decrease its activity because labeling non-specifically targets lysine residues that may be functionally important for augmin and/or tubulin binding (unpublished work). Thus, because we are working with the truncated augmin complex, which already binds more weakly to γ -TuRC than full length augmin, we do not expect that labeled γ -TuRC will robustly bind to truncated augmin in our reconstitution assay, at least without extensive troubleshooting of our γ -TuRC labeling strategy. However, we do hope to revisit these experiments at a later date in a publication more directly focused on the biochemical reconstitution.

9. The temperature at which the branching experiments are performed should be stated, because it is an important parameter influencing microtubule nucleation. The tubulin concentration is relatively high, and one may expect that microtubules nucleate spontaneously at this concentration unless the temperature is fairly low.

We thank the reviewer for pointing this out. The branching experiments were performed at 33°C to stabilize the GMPCPP MT seeds. We had included this information in the Methods section, under the heading "In vitro branching MT nucleation", but we have also added the temperature of the reaction to the legend of Figure 1 to make this important information more easily accessible to the reader.

10. The fluorescence image showing the augmin channel is presented using contrast and brightness settings that appear to suppress any background signal and strongly enhance the contrast. A gentler way of brightness and contrast adjustment would probably give a more realistic representation of the situation.

We thank the reviewer for this suggestion. We have reprocessed the data presented in Figure 1b and Figure S1b and matched the contrast of all channels to the Cy5 channel to provide a clearer representation of the branching reaction.

11. It would be useful to indicate in at least one of the figures where in the augmin structure the missing HAUS6 piece is located and indicate schematically how big it may be.

We thank the reviewer for this suggestion. We have revised Figure 2 (our model figure) to visualize where the C-terminus of Haus6 is located within the complex, as well as its approximate size. We have also added a cartoon to Figure S4a that schematizes the domain architecture of the complex, including the relative size of the C-terminus of Haus6.

12. The authors discuss a difference between the recently published model for the structure of the augmin complex (Gabel et al., Nat. Comms., 2022) and cite Fig. S4. However, no comparison between the two structures is shown in this figure. It would be useful to illustrate the difference in the position of the calponin homology domains in the two models in a figure more directly.

We thank the reviewer for their helpful suggestion. We have added a supplementary figure (Figure S7) that compares our structure with those from two recent publications reporting augmin structures: Gabel et al, as well as a second paper that we learned of immediately after submitting our manuscript, Zupa et al, Nat. Comms., 2022. In particular, we have added two inset panels that explicitly address the differences in CH domain position between the augmin structures. We hope that this will aid the reader in visualizing the conformational differences between these structures.

13. The discussion about augmin in different species with the evolutionary implications is interesting. But a discussion about what we can learn from the structure of augmin, as presented here, for augmin's function is largely missing.

We thank the reviewer for their helpful feedback. Although evolution of protein complexes remains a particular interest of ours, we have substantially revised the discussion, reworking paragraph 4 and adding a fifth paragraph, to emphasize the conclusions that can be drawn directly from our structure of *X. laevis* augmin.

Reviewer #3:

Suggestions for revision:

1. *In the abstract the authors claim that branching microtubule nucleation "exponentially amplifies microtubules" - this implies a defined quantitative relationship, but has this really been quantified to demonstrate that it is indeed "exponential"? If not strictly supported by evidence, I suggest changing the wording to a less specific term*

We agree; we have revised the language to be more accurate.

2. *Also in the abstract: "branching has been hindered by a lack of structural information" - shouldn't it be "the"?*

We have corrected this.

3. *Abstract and beyond: "allowed" implies that an object has authority, perhaps using "enabled" is more appropriate*

We have corrected this.

4. *The introduction seems under-referenced in places:*

"Critically, spindle MTs have to be continuously generated, as any given MT turns over within seconds, yet the spindle framework must often persist for up to an hour. Due to the spindle's central importance for viability, multiple partially redundant pathways are used to generate spindle MTs."

Each statement should be referenced and some examples of these multiple (redundant) pathways might also be helpful for a more general reader.

We thank the reviewer for pointing this out. We have added additional citations for these statements and expanded on the redundant pathways used for generating spindle MTs, providing two of the most prominent examples.

5. *Also, terms such as "spindle body", "spindle assembly checkpoint" may need some brief introduction or definition*

We thank the reviewer for this helpful feedback. We have altered the text to reduce jargon and make the concepts more accessible to the general reader.

6. *"with the bottom of the "h" sitting on the MT" - sure not "an MT"?*

We have corrected this.

7. *"In addition, lack of any identified structural homologs" - perhaps "the lack"?*

We have corrected this.

8. Results: *"Expanding upon our results with D. melanogaster augmin, we next sought to answer another major unknown question about the augmin complex" - "unknown question" sounds a bit pleonastic*

We have revised the text to reduce redundancy.

9. Discussion: *"This model has allowed us not only to identify the positions and orientations of all eight subunits, but also to incorporate prior experimental knowledge to locate the MT and γ -TuRC binding sites within the complex, establishing for the first time the overall structure of the branching MT organizing center" - better to avoid the statement of priority here, even if true (but probably not)*

We thank the reviewer for bringing this to our attention. This language was a carry-over from a much earlier draft. We have revised the text to avoid statements of priority.

10. *"Haus6 and Haus7 displayed, unexpectedly, a classic calponin homology fold at their N-termini." - why "unexpectedly"?*

We have revised to clarify our meaning.

11. *"This second MT binding mode may help solve the mystery of how a disordered attachment via the unstructured N-terminus of Haus8 is capable of orienting γ -TuRC stably relative to the pre-existing MT, and more generally how branching MT nucleation is capable of maintaining spindle polarity" - would not call it a second binding mode in this context, suggest rephrasing to something like:*

"The second, firm/fixed anchor point may explain how branching..." - would be nice to give this concept a bit more emphasis in the figure, if possible, maybe by including some more arrows and labels in Fig 5a?

We thank the reviewer for their helpful feedback. We have revised and expanded this part of the discussion into a new paragraph (paragraph 5) to better clarify this idea. We have also modified Figure 5a (now Figure 5c) to include additional arrows, highlighting the two MT-binding regions, as well as adding labels. We hope that this will help communicate this concept, which we view to be one of the main conclusions of our study.

12. *Have not found any reference to Fig 5c in the text - what is illustrated should be worth discussing, I guess?*

Thank you for bringing this oversight to our attention. We have revised the flow of the discussion, as well as the figure layout, and ensured that we include a specific reference to this panel (now Figure 5b).

13. *Not sure if Materials and Methods provide enough information to replicate the study. For example, haven't found how tubulin was labelled - procedure or the commercial source should be provided.*

Several sections were added to the Materials and Methods section to provide enough information to replicate this portion of the study.

14. *"Schrödinger" not "Schrodinger"*

We have corrected the typo.

15. *Fig 1b - the most important bits are strictly invisible when printed. On the screen it is better, but the panels are too small and contrast is not great, perhaps close-ups would be good as well as larger overviews.*

We modified Figure 1 to use close-up views, to enhance contrast and visibility. We additionally included the larger overviews in Figure S1. Moreover, as requested by another reviewer, we contrast-matched the channels to provide a more realistic depiction of the experiments.

16. *Fig 2 legend: "Final model of T-II and T-III into moderate resolution cryo-EM map" - missing "fitted"?*

We have revised the text to include the missing word.

17. *Fig 3 legend: top/bottom panels maybe better than "above/below"*

We have revised the legend to incorporate this terminology.

18. *Fig 5a legend: "The NEDD1 γ -TuRC adaptor binding site is predicted to be located with T-III (pink)" - should be "within"?*

We have corrected the typo.

19. *Fig S1: close-up views would be good if not included in the main figure*

As discussed in response to point #15, we have increased the size of the panels presented in the main text figure to increase interpretability. We have retained the uncropped images in the supplemental figure to allow comparison between MT number across experiment and controls.

REVIEWERS' COMMENTS

Reviewer #1 (Remarks to the Author):

Congratulations to the authors for their nice revision. We spotted only a few very minor errors:

- The authors use sp-cryoEM but don't explain the abbreviated term properly
- Discussion, paragraph 2, sentence 2 does not make sense

Reviewer #2 (Remarks to the Author):

The authors have addressed my comments satisfactorily, as well as the comments of the other reviewers, as far as I can see. The manuscript has been nicely improved. Some typos can still be found that are worth correcting.

Point-by-point:

Reviewer #1 (Remarks to the Author):

Congratulations to the authors for their nice revision. We spotted only a few very minor errors:

- The authors use sp-cryoEM but don't explain the abbreviated term properly

We have removed the abbreviation from the text

- Discussion, paragraph 2, sentence 2 does not make sense

We have corrected a typo in this sentence, changing the first word 'A' to 'All three.' We hope that has fixed the issue.

Reviewer #2 (Remarks to the Author):

The authors have addressed my comments satisfactorily, as well as the comments of the other reviewers, as far as I can see. The manuscript has been nicely improved. Some typos can still be found that are worth correcting.

We double checked through the manuscript for typos and corrected a spelling error ("Stramenopiles" not "Straminopiles") and removed a second period in the legend for Figure 1c. We also proof-read and made a few minor word changes to improve conciseness.